# THO Complex Subunit 7 Homolog Negatively Regulates Cellular Antiviral Response against RNA Viruses by Targeting TBK1

**DOI:** 10.3390/v11020158

**Published:** 2019-02-15

**Authors:** Tian-Sheng He, Tao Xie, Jing Li, Ya-Xian Yang, Changsheng Li, Weiying Wang, Lingzhen Cao, Hua Rao, Cynthia Ju, Liang-Guo Xu

**Affiliations:** 1Key Laboratory of Functional Small Organic Molecules, Ministry of Education and College of Life Science, Jiangxi Normal University, 99 Ziyang Avenue, Nanchang 330022, Jiangxi, China; hetiansheng361@163.com (T.-S.H.); tao.fanfan0103@gmail.com (T.X.); christie567@163.com (J.L.); xianzi3jd@163.com (Y.-X.Y.); Csl732@163.com (C.L.); windowandme@163.com (W.W.); clzclz1011@163.com (L.C.); rshua@126.com (H.R.); 2Department of Anesthesiology, McGovern Medical School, University of Texas Health Science Center at Houston, Houston, TX 77030, USA

**Keywords:** THOC7, MAVS signalosome, TBK1, cellular antiviral response

## Abstract

RNA virus invasion induces a cytosolic RIG-I-like receptor (RLR) signaling pathway by promoting assembly of the Mitochondrial antiviral-signaling protein (MAVS) signalosome and triggers the rapid production of type I interferons (IFNs) and proinflammatory cytokines. During this process, the pivotal kinase TANK binding kinase 1 (TBK1) is recruited to the MAVS signalosome to transduce a robust innate antiviral immune response by phosphorylating transcription factors interferon regulatory factor 3 (IRF3) and nuclear factor (NF)-κB and promoting their nuclear translocation. However, the molecular mechanisms underlying the negative regulation of TBK1 are largely unknown. In the present study, we found that THO complex subunit 7 homolog (THOC7) negatively regulated the cellular antiviral response by promoting the proteasomal degradation of TBK1. THOC7 overexpression potently inhibited Sendai virus- or polyI:C-induced IRF3 dimerization and phosphorylation and IFN-β production. In contrast, THOC7 knockdown had the opposite effects. Moreover, we simulated a node-activated pathway to show that THOC7 regulated the RIG-I-like receptors (RLR)-/MAVS-dependent signaling cascade at the TBK1 level. Furthermore, THOC7 was involved in the MAVS signalosome and promoted TBK1 degradation by increasing its K48 ubiquitin-associated polyubiquitination. Together, these findings suggest that THOC7 negatively regulates type I IFN production by promoting TBK1 proteasomal degradation, thus improving our understanding of innate antiviral immune responses.

## 1. Introduction

Cellular innate immune responses are initiated by the detection of “non-self” conserved molecular motifs of invading microbial pathogens, such as pathogen-associated molecular patterns [1], or “self” tissue damage-inducing molecules released by host necrotic cells, such as danger-associated molecular patterns [2]. Several host germline-encoded sensor molecules known as pattern recognition receptors were identified, including the first-discovered membrane-spanning Toll-like receptors (TLRs) [3], cytosolic RIG-I-like receptors (RLRs) [4], nod-like receptors (NLRs) [5], and newly described cytosolic DNA sensors [6]. RIG-I activation after cytosolic viral RNA detection induces homotypic caspase activation and recruitment domain (CARD)–CARD interactions through the CARD domains of RIG-I and a central antiviral adaptor protein known as MAVS, which is also referred to as VISA, IPS-1, and Cardif [7,8,9,10] and located in the mitochondria. The mitochondria provide a molecular platform for transducing antiviral signals by promoting the assembly of a MAVS signalosome [11]. The MAVS signalosome then recruits the important kinase TBK1 to activate the transcription factors IRF3 and NF-κB, followed by their nuclear translocation and high production of type I interferons (IFNs) and proinflammatory cytokines [12].

TBK1, a member of a noncanonical IKK family of serine/threonine kinases, is ubiquitously expressed in various cells and acts as a key node protein for regulating IRF3 activity and downstream type I IFN production [13]. TBK1 is suggested to phosphorylate three vital adaptor proteins, MAVS, Stimulator of interferon genes protein (STING), and TIR domain-containing adapter protein inducing IFN-beta (TRIF), in response to stimulation by viral RNA, viral DNA, and bacterial lipopolysaccharide (LPS), respectively [14]. TBK1 also phosphorylates the downstream transcription factor IRF3 recruited by the phosphorylated adaptor proteins because of its increased proximity. Because TBK1 plays a critical role in cytoplasmic antiviral immunity, its activity is suggested to be tightly regulated by different regulators, particularly efficient inhibitors, through multiple pathways to prevent an excessive immune response. Many studies show that various molecules regulate TBK1-mediated signaling through post-translational modification, including phosphorylation by Glycogen synthase kinase-3 beta (GSK3β) [15], Phosphatidylinositol 3,4,5-trisphosphate 5-phosphatase 1 (SHIP-1) [16], and Protein phosphatase 1B (PPM1B) [17], and ubiquitination by Mind bomb homolog (MIB)1/2 [18], Suppressor of cytokine signaling 3 (SOCS3) [19], E3 ubiquitin-protein ligase NRDP1 [20], RING finger protein 11 (RNF11)- Tax1-binding protein 1 (TAX1BP1) [21], and NLR family pyrin domain containing 4 (NLRP4)-Deltex E3 ubiquitin ligase 4 (DTX4) [22]. Schmitz reported that SUMOylation is a novel modification of TBK1 and that a SUMO acceptor site K694 located in the C-terminal coiled-coil domain of TBK1 is responsible for the binding of TBK1 to its adaptor proteins TRAF family member-associated NF-κB activator (TANK), Neutrophil-activating protein 1 (NAP1), and TBK1 binding protein 1 (TBKBP1) [23]. Although several TBK1 inhibitors were identified, it is important to identify more efficient TBK1 inhibitors to determine the mechanisms underlying its antiviral response and homeostatic activity in the host.

THO complex subunit 7 homolog (THOC7), a member of the THO transcription elongation complex, is required for the efficient nuclear export of mRNA [24]. THOC7 is also referred to as NIF3L1-binding protein 1 because it is the cytoplasmic interaction partner of putative transcriptional repressor NIF3L1 [25]. The THO complex forms a transcription/export (TREX) complex by interacting with DEAD-box protein RNA helicases DDX39B, SARNP/CIP29, and CHTOP in an ATP-dependent manner [24,26]. Whitehouse showed that the TREX complex is required for nuclear transport of Kaposi’s sarcoma-associated herpesvirus (KSHV) intron-less mRNA and production of infectious viral particles [26]. In the present study, we determined the role of the TREX complex component THOC7, which was identified as a novel TBK1-interacting protein by yeast two-hybrid screening, in the RLR signaling pathway induced by RNA viruses. THOC7 overexpression inhibited Sendai virus (SeV)- or polyI:C-induced IFN-β promoter activation, and *THOC7* knockdown strengthened IRF3 activation and IFN-β production. THOC7 interacted with TBK1 and was increased after viral infection. Subsequently, THOC7 promoted TBK1 degradation through a ubiquitin-dependent degradation system. These findings indicate that THOC7 is a novel TBK1 inhibitor that negatively regulates innate antiviral immunity to maintain immune homeostasis.

## 2. Materials and Methods

### 2.1. Cells, Viruses, Antibodies, and Reagents

HEK293T cells and MCF7 cells were provided by Dr. Hong-Bing Shu (Wuhan University, China) and cultured in Dulbecco’s modified Eagle’s medium (Gibco; Grand Island, NY, USA) supplemented with 10% fetal bovine serum (Gibco), penicillin (100 U/mL; Solarbio, Beijing, China), and streptomycin (100 U/mL; Solarbio) at 37 °C in an incubator with a 5% CO_2_ atmosphere. Sendai virus (SeV) was generated as described previously [7,27]. Lipofectamine™ 3000 transfection reagent was purchased from Thermo Fisher Scientific (Waltham, MA, USA). MG132 (5 μM; InvivoGen, San Diego, CA, USA) and cycloheximide (CHX, 20 μM; InvivoGen, USA) were added in medium to evaluate the degradation of TBK1. Mouse monoclonal antibodies against HA/FLAG tag and Myc tag were purchased from Sigma (St. Louis, MO, USA) and Santa Cruz Biotechnology (Dallas, TX, USA), respectively. Horseradish peroxidase (HRP)-conjugated anti-mouse and anti-rabbit IgG antibodies were purchased from Bio-Rad (Hercules, CA, USA) and Cell Signaling Technology (Danvers, MA, USA), respectively. Antibodies against the RLR signaling pathway components (sampler kit #8348), including IRF3, phosphorylated IRF3 (Ser396) (p-IRF3), and TBK1, were purchased from Cell Signaling Technology. Low-molecular-weight polyI:C was purchased from Invivogen (San Diego, CA, USA).

### 2.2. Plasmids

Luciferase reporter plasmids containing an IFN-sensitive response element (ISRE), NF-κB, or IFN-β promoter conjugated to the firefly luciferase reporter gene and mammalian expression vectors expressing RLR signaling pathway components, including RIG-I and its mutant RIG-I-N (deleted C-terminal repressor domain and DECH-box helicase domain), MAVS, TBK1, IKKε, IRF3 and its point mutant IRF3-5D (active form of IRF3), and ubiquitin and its mutant K48 or K63 ubiquitin, were prepared as described previously [7,27]. Human *THOC7* (h*THOC7*) was cloned into a cytomegalovirus promoter-based mammalian expression vector along with a sequence encoding an N-terminal HA, FLAG, or Myc tag using standard molecular biology techniques. THOC7-specific siRNA constructs were generated by cloning double-stranded oligonucleotides corresponding to the h*THOC7* target sequence into an RNA interference (RNAi) vector pSuper.retro (OligoEngine, Seattle, WA, USA) according to the manufacturer’s protocol. The following target sequences were designed for h*THOC7* cDNA: *THOC7*-RNAi#1, 5′-GGAAACAGTTTCATGTTCT-3′; *THOC7*-RNAi#2, 5′-CCAGACAGGCATGAGACAT-3′; and *THOC7*-RNAi#3, 5′-TCTTCTTAGTACCATCCAT-3′.

### 2.3. Yeast Two-Hybrid Screening Assay and Dual-Luciferase Reporter Assay

A full-length *TBK1* fragment was cloned into the pGBT9 vector containing a GAL4 DNA-binding domain (amino acids, 1–147), and the resulting pGBT9-TBK1 was used as a bait for performing yeast two-hybrid screening of a human 293T cDNA library, which was fused with a GAL4 DNA activation domain (amino acids, 768–881). Large-scale screening was performed as described previously [27]. Positive clones were selected by culturing the cells in nutrient-deficient culture medium (Try^−^, Leu^−^, and His^−^), then sequenced at BGI (Shenzhen, China). Data were analyzed using BLAST. Dual-luciferase reporter assay was performed by co-transfecting the 293T cells with 100 ng ISRE-, NF-κB promoter-, or IFN-β promoter-containing luciferase reporter construct, 50 ng pRL-TK (*Renilla* luciferase) plasmid, and different doses of pRK5-THOC7 (0.1, 0.2, 0.4, and 0.8 μg) or HAUS8-specific siRNAs (0.5 μg) using a standard calcium phosphate precipitation method as described previously [7,27]. The cells were then treated with or without SeV for 10 h and harvested at 20 h after transfection. The luciferase activity of whole-cell lysates was measured with a GloMax™ luminometer (Promega, Madison, WI, USA) and dual-luciferase assay kit (Promega). Relative luciferase activity was normalized based on the luciferase activity of the pRL-TK plasmid as a control. The experiment was repeated at least three times.

### 2.4. Coimmunoprecipitation, Immunoblotting, and Native PAGE Assays

To perform transient transfection and coimmunoprecipitation assays, the 293T cells (density, ~6 × 10^6^) were plated in 100 mm dishes and transfected with different expression vectors using the standard calcium phosphate precipitation method. At 20 h after transfection, the cells were collected and lysed using 1 mL Triton X-100 lysis buffer. For the immunoprecipitation assay, 800 μL cell lysate was incubated overnight at 4 °C with ~30 μL protein A/G-Sepharose beads (GE Healthcare, Piscataway, NJ, USA) and 0.3 μg of the indicated antibodies. Next, the Sepharose beads were washed three times with 1 mL lysis buffer containing 1 M NaCl and 1 mL lysis buffer lacking NaCl for 20 min. The precipitates obtained were boiled and analyzed by SDS-PAGE as described previously [7,27]. Protein dimerization was evaluated in a native PAGE assay. For this assay, cellular extracts were dissolved in native PAGE sample buffer (62.5 mM Tris-Cl [pH 6.8], 15% glycerol, and 1% deoxycholate) without boiling [28,29], and native PAGE was performed in an ice bath as described previously [30]. Proteins were visualized by electrogenerated chemiluminescence (ECL) imaging in the Gel Doc XR Gel Documentation System (Bio-Rad, CA, USA) according to the manufacturer’s instructions. Quantification of the immunoblot was performed with Image J software (NIH, Bethesda, MD, USA) after repeating the experiment three independent times.

### 2.5. RNA Purification and Fluorescent Quantitative PCR

Total RNA was isolated from transfected cells using Eastep^®^ Super RNA extraction kit (Promega) according to the manufacturer’s instructions. Next, cDNA was synthesized from 1 μg total RNA using the GoScript™ reverse transcription kit (Promega). Quantitative PCR (qPCR) was performed using Eastep^®^ qPCR SYBR Master Mix (Promega) and 7500 Real-Time PCR System (Applied Biosystems, Foster City, CA, USA). Relative mRNA levels of the target genes were normalized to the mRNA level of the β-actin gene and calculated by the 2^−ΔΔCq^ method [31]. The sequences of primers against the target genes were as follows: h*THOC7* forward, 5′-CCGTGACTGACGACGAAGTT-3′; h*THOC7* reverse, 5′-CAGCATACGTTGGTACTGGC-3′; IFN-β gene forward, 5′-CTAACTGCAACCTTTCGAAGC-3′; IFN-β gene reverse, 5′-GGAAAGAGCTGTAGTGGAGAAG-3′; ISG56 gene forward, 5′-TCATCAGGTCAAGGATAGTC-3′, ISG56 reverse, 5′-CCACACTGTATTTGGTGTCTAGG-3′; β-actin gene forward, 5′-GTCGTCGACAACGGCTCCGGCATG-3′; and β-actin gene reverse, 5′-ATTGTAGAAGGTGTGGTGCCAGAT-3′.

### 2.6. Statistical Analysis

All data shown in the histograms were analyzed with GraphPad Prism software (version 7.0; GraphPad Software, Inc., La Jolla, CA, USA) and are presented as the mean and standard deviation of at least three independent experiments. Statistical significance of differences between two groups were analyzed using an unpaired Student’s *t*-test and those of differences among multiple groups were analyzed by one-way analysis of variance with Tukey’s post hoc analysis. *p* < 0.05 was considered as statistically significant. The different number of asterisks indicates the degree of significance with respect to the *p* values.

## 3. Results

### 3.1. THOC7 Overexpression Negatively Regulates Type I IFN Production

To identify potential proteins that regulate type I IFN production, we performed a large-scale yeast two-hybrid screening assay using the full-length TBK1 as a bait protein. The yeast two-hybrid screening assay identified THOC7 (GenBank accession number: NM_025075.3) as a candidate protein that interacted with TBK1, which was confirmed by gene sequencing and BLAST analysis. The results of transient transfection and coimmunoprecipitation assays validated that THOC7 interacted with TBK1 in 293T cells (Figure 1A). Next, we systematically investigated whether THOC7 regulated IFN-β signaling. A small-scale dual-luciferase reporter assay was performed to determine whether THOC7 regulated RNA virus- or polyI:C-induced type I IFN signaling (Figure 1B). The results of the dual-luciferase reporter assay showed that THOC7 markedly suppressed polyI:C- or SeV-induced ISRE and IFN-β promoter activation. Notably, THOC7 overexpression markedly suppressed TBK1-induced and the combination of TBK1-, polyI:C-, and SeV-induced ISRE and IFN-β promoter activation.

Next, systematic dual-luciferase report assays were performed to detect type I IFNs triggered by SeV. The results of these assays showed that THOC7 suppressed the activity of the IFN-β promoter (Figure 1C) in 293T cells and MCF7 cells in a dose-dependent manner. Moreover, THOC7 inhibited IFN-stimulated response element (ISRE)-luciferase activity (Figure 1D) and NF-κB promoter activity induced by SeV. Because IRF3 acts as a vital transcription factor to initiate IFN transcription, phosphorylation and dimerization of IRF3 are the hallmarks of IFN production and the antiviral response. Therefore, we measured IRF3 dimerization by performing a native PAGE assay and determined the IRF3 phosphorylation level by performing an immunoblotting assay with the anti-p-IRF3 antibody. Consistent with our previous results, we observed that THOC7 overexpression inhibited SeV-induced activation of IRF3 dimerization and phosphorylation (Figure 1F). Additionally, the RT-PCR results showed that THOC7 overexpression resulted in lower expression of IFN-β mRNA and interferon-stimulated gene ISG56 mRNA induced by viruses. Together, these results indicate that THOC7 negatively regulates the innate immune response against RNA viruses.

### 3.2. THOC7 Knockdown Upregulates Type I IFN Production

To further investigate whether THOC7 regulated SeV-induced IFN production under physiological conditions, we examined the effects endogenous *THOC7* knockdown. For this, three pSUPER.retro-based RNAi expression constructs (*THOC7*-RNAi#1, *THOC7*-RNAi#2, and *THOC7*-RNAi#3) targeting different regions of the h*THOC7* mRNA were constructed, and knockdown efficiency was monitored. The three *THOC7*-RNAi constructs remarkably downregulated the expression of transiently transfected *THOC7* (Figure 2A) and endogenous *THOC7* (Figure 2B, left panel). Moreover, siRNA-induced downregulation of *THOC7* expression increased SeV-induced IFN-β gene transcription (Figure 2B, right panel).

Next, we examined IRF3 dimerization by performing a native PAGE assay and IFN-β activation by performing a dual-luciferase reporter assay. Our results showed that *THOC7* knockdown significantly increased IRF3 dimerization and SeV-induced ISRE and IFN-β promoter activation (Figure 2C,D). Notably, the *THOC7*-RNAi#3 sequence was associated with the highest knockdown efficiency and highest IFN-β activation among the three *THOC7*-RNAi sequences. Therefore, this sequence was selected for subsequent analysis. Next, we performed a time-dependent SeV-induced IRF3 activation assay and observed an obvious increase in IRF3 dimerization compared to that in control cells after activation of RLR signaling pathway (Figure 2E). Moreover, *THOC7* knockdown enhanced SeV-induced IRF3 phosphorylation (Figure 2F). These data confirm that THOC7 negatively regulates type I IFN production.

### 3.3. THOC7 Regulates the RLR Signaling Pathway by Targeting TBK1

Upon viral infection, RIG-I detects viral RNA through its central DEAD box helicase/ATPase domain and is activated after exposure of its CARD domain to the virus [32,33]. Activated RIG-I recruits MAVS through homotypic CARD–CARD interactions on the mitochondria, which serve as the platform for inducing an innate immune response by promoting the assembly of the MAVS signalosome [12]. Next, TBK1 is recruited to the MAVS signalosome and phosphorylates MAVS [34]. IRF3 then interacts with the phosphorylated MAVS and is phosphorylated by TBK1 because of the proximity of these two proteins. The activated IRF3 forms a homodimer and undergoes nuclear translocation to induce type I IFN production. To determine the mechanisms underlying the regulation of RNA virus-induced innate immune response by THOC7, we established a model simulating a node-activated pathway by transiently transfecting the 293T cells with vectors expressing the activated forms of the proteins involved in this pathway (Figure 3A). Activation of a single node only affects its downstream signaling but has a limited effect on its upstream signaling. The cells were transfected with vectors expressing activated RIG-I and IRF3 mutant proteins, namely, RIG-I-N (only containing the CARD domain and deleted repressor domain) and IRF3-5D (analogous to p-IRF3, phosphate mimic by aspartic acid), respectively, and with the vectors expressing the other activated proteins, namely, MAVS, TBK1, and IKKε, involved in the RLR signaling pathway. Overexpression of these proteins by transient transfection could simulate its activation state and triggered downstream signaling.

THOC7 overexpression significantly attenuated RIG-I-N-, MAVS-, TBK1-, and IKKε-mediated ISRE or IFN-β promoter activation but did not attenuate IRF3-5D-mediated ISRE or IFN-β promoter activation (Figure 3B). In contrast, *THOC7* knockdown increased RIG-I-N-, MAVS-, TBK1-, and IKKε-mediated ISRE or IFN-β promoter activation but did not increase IRF3-5D-mediated ISRE or IFN-β promoter activation (Figure 3C). Notably, THOC7 mainly acted at the TBK1 level, but not at the IKKε level, as indicated by the degree of the effect, despite the redundancy in the functions of these proteins. These results suggest that THOC7 negatively regulates the RLR antiviral signaling pathway by targeting TBK1.

### 3.4. THOC7 Is Involved in MAVS Signalosome and Promotes the Proteasomal Degradation of TBK1

The RLR signaling pathway is propagated by assembly of the MAVS signalosome [11]. Therefore, we investigated whether THOC7 was involved in forming the MAVS signalosome. THOC7 interacted with MAVS, TBK1, IKKε, and IRF3 and, notably, the interaction was decreased with MAVS, but increased with TBK1 after virus infection (Figure 4A), indicating that THOC7 have a significant function by targeting TBK1 in the antiviral response. Moreover, we observed that THOC7 overexpression decreased TBK1 expression. Therefore, we examined the abundance of TBK1 in the presence of THOC7. Our results showed that THOC7 overexpression induced degradation of exogenous and endogenous TBK1 in a dose-dependent manner (Figure 4B). Additionally, we examined the effect in the presence and absence of THOC7 in MCF7 cells and found a weak downregulation of TBK1 by THOC7 expression, but obvious effect upon virus infection. (Appendix A). We predicted that THOC7-induced TBK1 degradation was driven by the ubiquitin–proteasome system. To verify the mechanism, MG132 and CHX were supplemented in the TBK1 degradation system without newly synthesized TBK1 protein. The results showed that MG132 efficiently restored the TBK1 protein level in the presence of THOC7 (Figure 4C). Subsequently, we observed that THOC7 extensively promoted the ubiquitination of TBK1 compared to that of RIG-I (Figure 4D). Next, we examined the specific type of ubiquitin involved in TBK1 degradation by transfecting the cells with vectors expressing two ubiquitin mutants, particularly K48 and K63 ubiquitin. Our results suggested that THOC7 increased K48 ubiquitin-associated TBK1 ubiquitination and attenuated K63 ubiquitin-associated TBK1 ubiquitination (Figure 4E). Collectively, these results suggest that THOC7 is involved in the MAVS signalosome and promotes the proteasomal degradation of TBK1.

## 4. Discussion

Virus invasion rapidly activates innate immunity by inducing the production of type I IFNs that robustly eliminate invading viruses and protect the host. However, activated innate immunity should be negatively regulated to efficiently prevent the induction of autoimmune diseases such as systemic lupus erythematous and Aicardi–Goutieres syndrome [35]. TBK1 plays a critical role as an integrator and inducer of RIG-I–MAVS, cGAS–STING, and TLR4–TRIF signaling pathways in response to infection by RNA viruses, DNA viruses, and bacterial LPS, respectively. Therefore, TBK1 is considered a promising target for preventing autoimmune diseases. Numerous studies have characterized the mechanisms regulating TBK1 activity, including ubiquitination, phosphorylation, modulation of TBK1 kinase activity, and prevention of functional TBK1-containing complex formation [36].

The THO complex was originally identified in *Saccharomyces cerevisiae* as a five-protein (THOC1, THOC2, THOC3, MFT1, and THP2) complex involved in mitotic recombination, transcription elongation, and mRNA nuclear export [37]. In higher eukaryotes, including humans, the THO complex contains three additional novel proteins, THOC5, THOC6, and THOC7, which have no apparent homologs in yeast [26] and likely perform more complicated functions. A larger complex known as TREX, which performs important functions in many cellular events and viral infection, contains the THO complex, RNA helicase DDX39B, and mRNA export NXF/TAP-adaptor protein THOC4 [38]. The TREX complex is used by different viruses, including herpesvirus [39], avian influenza virus [40], and murine leukemia virus [41], to complete their lifecycle. Moreover, the complete TREX complex is required for the mRNA nuclear export and replication of KSHV [42], a human gamma herpesvirus that is primarily associated with lymphoma, Kaposi’s sarcoma, and multicentric Castleman’s disease [43]. One study reported that an early master switch protein known as replication and transcription activator of KSHV downregulates the expression of TLR adaptor MyD88 [43]. In the present study, we found that the TREX complex protein THOC7 is an important regulator of the RLR antiviral signaling pathway.

Our results suggest that THOC7 negatively regulates the cellular antiviral response by targeting TBK1. First, we found that THOC7 overexpression suppressed SeV-induced IRF3, NF-κB promoter, and IFN-β promoter activation, as indicated by the results of the luciferase reporter assays, and that *THOC7* knockdown exerted opposite effects. Next, we found that THOC7 overexpression inhibited RIG-I-, MAVS-, and TBK1-mediated IRF3 and IFN-β promoter activation but did not inhibit IRF3-mediated IRF3 and IFN-β promoter activation, which is consistent with the results of *THOC7* knockdown experiments. These data indicate that THOC7 regulates the RLR signaling pathway upstream of IRF3. Additionally, we found that THOC7 is involved in the MAVS signalosome and promotes the polyubiquitination of TBK1. Together, these results suggest that THOC7 negatively regulates type I IFN production by promoting TBK1 degradation. Additionally, THOC7 suppresses NF-κB activation (Figure 1E) likely by targeting canonical IKKs (IKKα/IKKβ/IKKγ), which show sequence homology with TBK1/IKKε and regulate NF-κB activation. Additional studies are needed to determine how THOC7 regulates NF-κB signaling and inflammation.

The TREX complex plays a major role in mRNA processing in the nucleus [44]. Because THOC7 does not contain a typical nuclear localization signal (NLS) and is mainly localized in the cytoplasm [45], the nuclear localization of THOC7 depends on its direct interaction with FMS-interacting protein [45]. However, the specific functions of THOC7 in the cytoplasm have not been reported. In the present study, we evaluated the novel role of THOC7 in cytoplasmic antiviral signaling, which was involved in the MAVS signalosome and inhibited type I IFN production by promoting TBK1 degradation. Thus, we identified a novel function of THOC7 in the cytoplasm and provide a more complete model of the RLR signaling pathway. This is the first study to demonstrate the novel role of cytosolic THOC7 in innate antiviral immunity and highlight important mechanisms underlying the negative regulation of TBK1. This provides insights for suppressing type I IFN signaling and maintaining immune homeostasis.

## 5. Conclusions

In this study, we identified a novel role for THOC7 in cellular antiviral signaling against RNA viruses, which was involved in the MAVS signalosome and promoted TBK1 degradation by increasing its K48 ubiquitin-associated polyubiquitination.

## Figures and Tables

**Figure 1 viruses-11-00158-f001:**
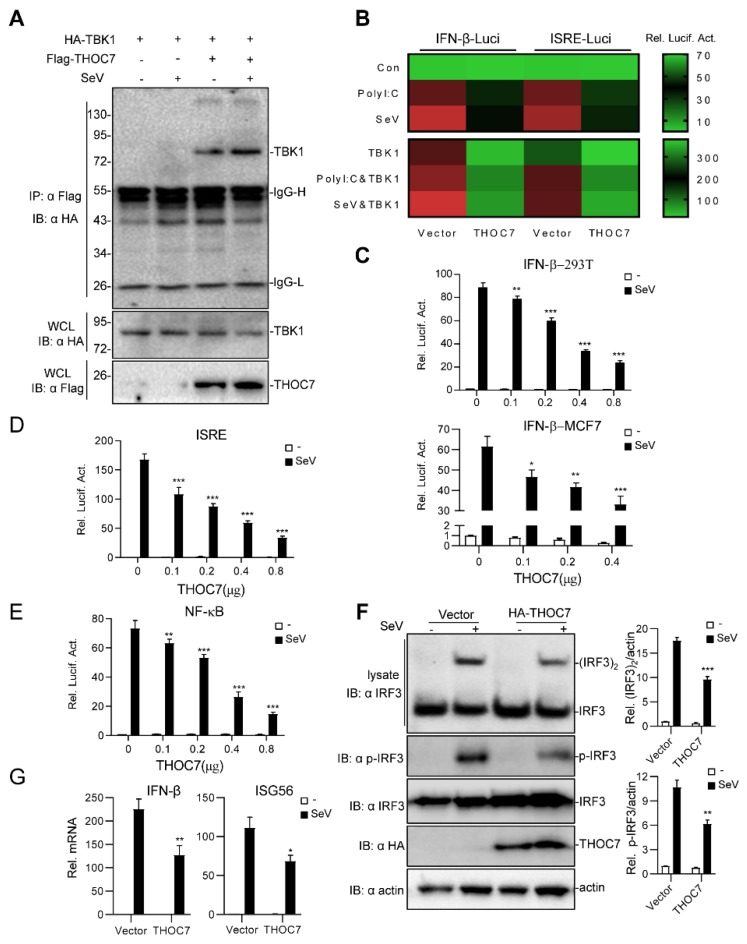
Overexpression of THOC7 negatively regulates the production of type I interferons (IFNs). (**A**) Interaction of THOC7 with TBK1 in a mammalian overexpression system. 293T cells were transfected with mock control, Flag-THOC7, and HA-TBK1 (10 μg each), followed by treatment with Sev or not for 10 h. At 24 h post-transfection, co-immunoprecipitation was performed with anti-HA beads and immunoblotting analysis with anti-Flag antibodies. (**B**) Small-scale screening of THOC7 functions in regulating SeV- or polyI:C-induced type I IFN signaling. 293T cells were seeded into 24-well plates and transfected with a luciferase reporter gene carrying the ISRE or IFN-β promoter (100 ng/well), pRL-TK (50 ng/well) and THOC7, followed by infection with SeV or transfection with polyI:C/TBK1 for 12 h. Relative luciferase activity levels were arbitrarily set to 0.1 (green). (**C**–**E**) THOC7 inhibited IFN-β promoter, ISRE, and NF-κB luciferase activation in a dose-dependent manner. Similar luciferase assay was performed, except with increasing amounts of the expression vector for THOC7. (**F**) THOC7 significantly reduced the phosphorylation and dimerization of IRF3. 293T cells were seeded into 6-well plates and transfected with HA-THOC7 (4 μg). After transfection for 12 h, the cells were treated with SeV or not for 10 h. Lysates were analyzed by native PAGE or SDS-PAGE. Quantification of western blotting bands from three independent experiments was performed with Image J software. (**G**) THOC7 inhibits IFN-β gene and ISG56 transcription. Similar transfection as described for (**F**) was performed. The mRNA levels were measured by q-PCR. Error bars indicate SD. *n* = 3. *, *p* < 0.05; **, *p* < 0.01; ***, *p* < 0.001; ns, no significant difference.

**Figure 2 viruses-11-00158-f002:**
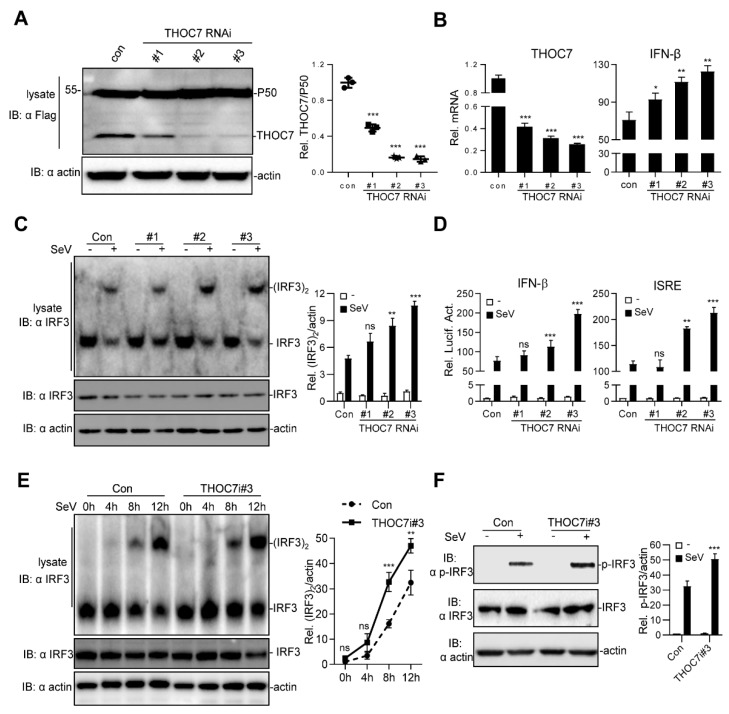
Knockdown of THOC7 leads to up-regulation of type I IFNs. (**A**) Immunoblot analysis following knockdown of transfected THOC7 in 293T cells transfected with THOC7-specific RNAis (2 μg), Flag-THOC7 plasmid (1 μg), and Flag-P50 (0.1 μg, as an internal control protein). (**B**) Real-time PCR of knockdown of exogenous THOC7 (left panel) or IFN-β mRNA (right panel) in 293T cells transfected with control RNAi or RNAis against THOC7 (4 μg). (**C**) Native PAGE analysis of IRF3 dimerization in 293T cells transfected with THOC7-specific RNAis (4 μg) and treated with SeV for 12 h. (**D**) Luciferase assays of 293T cells transfected with ISRE or IFNβ luciferase report plasmid (100 ng/well) with pRL-TK (50 ng/well), as well as THOC7 RNAi or control (0.5 μg/well). (**E**) IRF3 dimerization analysis of THOC7 RNAi#3 in gradient times of SeV treatment. Similar experiments were performed as (**C**) except with an increased time of SeV treatment. (**F**) Effects of knockdown of THOC7 on SeV-induced phosphorylation of IRF3. 293T cells were transfected with control RNAi or THOC7 RNAi#3 and infected with SeV for 10 h, followed by immunoblotting analysis with the indicated antibodies. Quantification of western blotting bands (**A**,**C**,**E**,**F**) from three independent experiments was performed with Image J software. Error bars indicate SD. *n* = 3. *, *p* < 0.05; **, *p* < 0.01; ***, *p* < 0.001; ns, no significant difference.

**Figure 3 viruses-11-00158-f003:**
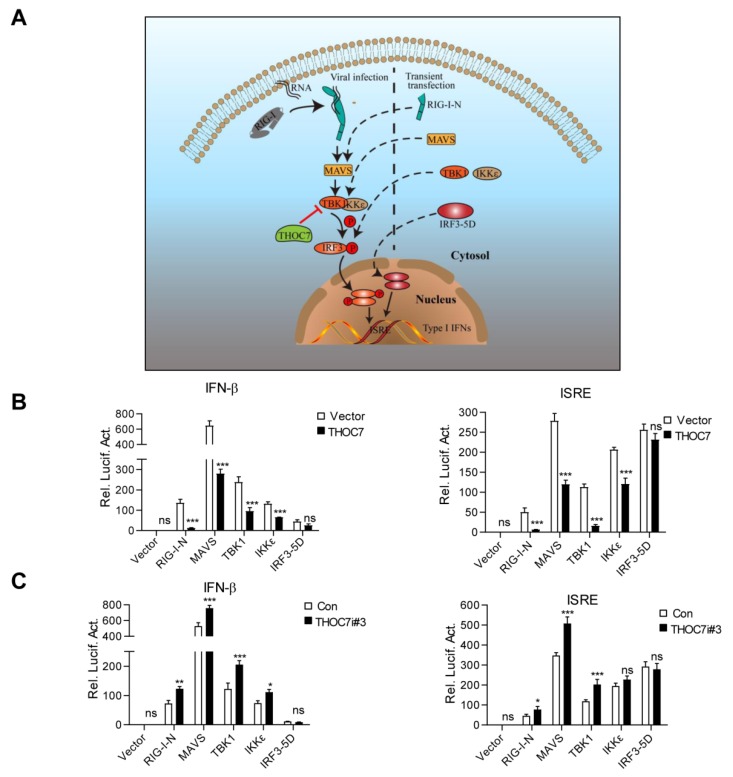
THOC7 regulates the RLR signaling pathway by targeting TBK1. (**A**) A simulation model of node-activated pathway for the role of THOC7 in regulating the RLR signaling pathway. Downstream signaling was activated by transfection of the activated form of signaling pathway molecules containing RIG-I-N, MAVS, TBK1, IKKε, and IRF3-5D. (**B**,**C**) Effects of THOC7 overexpression or THOC7 knockdown on IFNβ or ISRE activation mediated by various activated forms of signaling molecules. 293T cells were seeded into 24-well plates and transfected with IFN-β or ISRE luciferase reporter (100 ng/well), pRL-TK (50 ng/well), THOC7 expression vector (0.5 μg, B), or THOC71 RNAi#3 constructs (0.5 μg, C) together with indicated activated forms of signaling molecules. Cells were harvested and analyzed by dual-luciferase reporter assay. After transfection for 24 h, the relative luciferase activities were normalized based on pRL-TK control activities. Error bars indicate SD. *n* = 3. *, *p* < 0.05; **, *p* < 0.01; ***, *p* < 0.001; ns, no significant difference.

**Figure 4 viruses-11-00158-f004:**
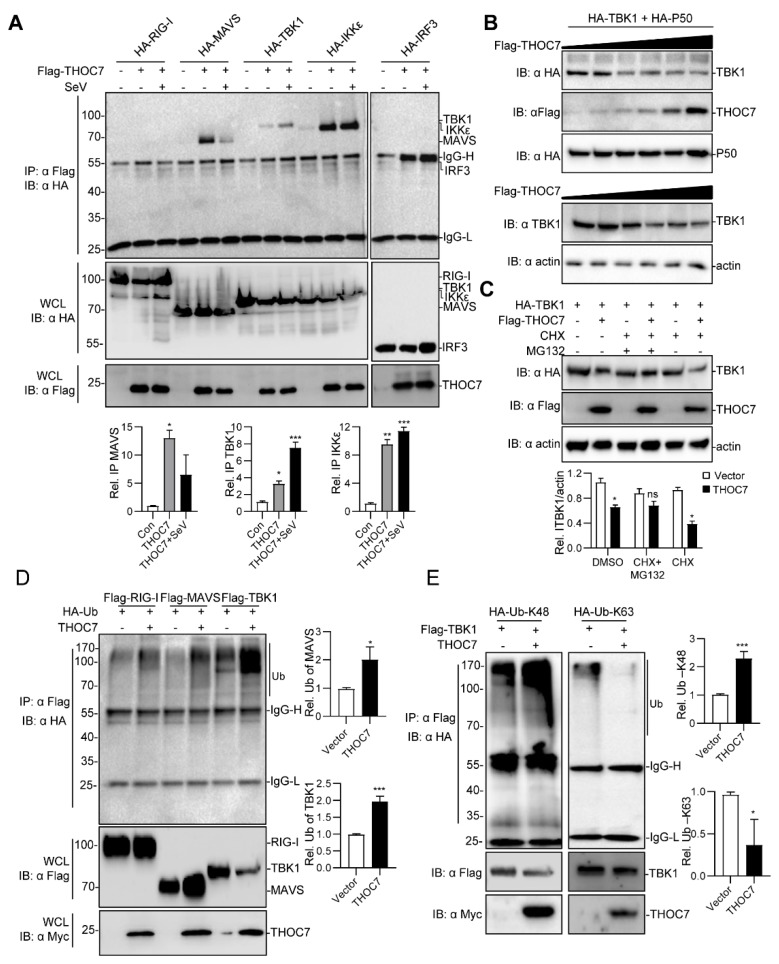
THOC7 is involved in the (mitochondrial antiviral signaling (MAVS) signalosome and drives proteasomal degradation of TBK1. (**A**) THOC7 was involved in forming the MAVS signaling complex. 293T cells were seeded into 100-mm dishes and transfected with Flag-tagged THOC7 or empty control plasmids (10 μg) and HA-tagged RLR signaling molecules (10 μg) as indicated. After 12 h of transfection, the cells were treated with or without SeV for 12 h. Co-immunoprecipitation and immunoblot analysis was performed with the indicated antibodies. (**B**) THOC7 promoted degradation of exogenous and endogenous TBK1 in a dose-dependent manner. 293T cells were seeded into 6-well plates and transfected with increasing amounts of Flag-THOC7 (0, 0.1, 0.2, 0.4, 0.8, 1.2 μg), and HA-TBK1 (2 μg). Twenty-four hours after transfection, immunoblot analysis was performed with the indicated antibodies. (**C**) MG132 restored the TBK1 protein level in the presence of THOC7. 293T cells were seeded into 6-well plates and transfected with Flag-THOC7 (2 μg) and HA-TBK1 (1.5 μg). MG132 and CHX were added after 6 h of transfection and the cells were harvested for analysis after 24 h. (**D**,**E**) THOC7 increased the K48-linked ubiquitination of TBK1. 293T cells were seeded into 100-mm dishes and transfected with the indicated plasmids (8 μg each). Similar co-immunoprecipitation and immunoblotting experiments were performed with the indicated antibodies. Quantification of western blotting bands (**A**,**C**,**E**,**F**) from three independent experiments was performed with Image J software. Error bars indicate SD. *n* = 3. *, *p* < 0.05; **, *p* < 0.01; ***, *p* < 0.001; ns, no significant difference.

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
