# Peer review of "THO Complex Subunit 7 Homolog Negatively Regulates Cellular Antiviral Response against RNA Viruses by Targeting TBK1"

_viruses, 2019, doi:10.3390/v11020158_

Round 1
Reviewer 1 Report
In the presented manuscript, the team around Liang-Guo Xu report on their discovery of a novel negative regulator of cell-intrinsic antiviral RIG-I-signaling. By means of a yeast two-hybrid screen, the authors identified THOC7, a component of the TREX complex, to interact with TBK1. In their very concise manuscript, they can validate the interaction by co-immuno-precipitation of the two proteins. They can further demonstrate that overexpression of THOC7 in 293T cells dose-dependently decreases antiviral singling and IFN-beta production, and, vice versa, knock-down of endogenous THOC7 increased the antiviral response. The authors show that this interference with the RIG-I pathway occurs at the level of TBK1 and IKKe, which is not too surprising, given its interaction with this kinase. Lastly, they provide data suggesting that THOC7 overexpression leads to increased K48-ubiquitylation of TBK1 and subsequent proteasomal degradation of TBK1, and speculate that this might be the mechanism of inhibition.
Overall, the study deals with an interesting topic, as in particular negative regulation of this pathway is not yet understood. The manuscript is, as said, very concise, with only four figures and a very compact text. This is nothing bad per se, and it is written in a very straight-forward manner, so that following the flow of experiments is easy. There are, however, a few typos and grammatical errors that should be taken care of. Figures are, by and large, well prepared and comprehensible. I would still like to raise a few issues that I feel have to be addressed prior to publication.
- General comment 1: from my own extensive experience, I know how difficult it is to get promoter reporter assays and also qRT-PCR measurements upon SeV infection to be exactly reproducible. SDs of the data shown throughout the manuscript are so tiny that I would just like to ask the authors to double-check and confirm that all data really is from three INDEPENDENT experiments and error bars represent SD.
- General comment 2: it would be advisable to check the THOC7 effect in at least one different cell line. As 293T cells seem to have very low expression of THOC7, the authors might want to try a cell line with higher expression, such as U2-OS or CAPAN-2 (check https://www.proteinatlas.org/ENSG00000163634-THOC7/cell#rna)
- Figure 1F: native PAGE is not convincing- where is the monomeric IRF3? Compare to fig. 2C! Moreover, the interpretation of this single experiment (line 179, 193, abstract, discussion) is not backed by this blot (“THOC7 significantly reduced phosphorylation and dimerization”). This experiment should be repeated and quantified (n=3).
- Figure 2: panel B seems to show the quantification of the blot in panel A across three repetitions. This is great and exactly what I would suggest to show also for the other blots (see comment above). However, there is not legend text for panel B- please also state how western blots were quantified!
- Figure 2: please explain why endogenous protein was not looked at? Was protein expression too low? There should be an antibody available!
- Figure 2E: again, monomeric IRF3 is not showing up, is there an explanation for this? Furthermore, please repeat to a total of n=3 and include error bars in the quantification (panel E right).
- Figure 2F: quantify, see above!
- Figure 3: see “general comment 1” (n and SD)
- Figure 4A: I cannot follow the interpretations of this experiment! To me, it seems very clear that THOC7 constitutively interacts with MAVS and IKKe, possibly a little bit also with TBK1. After virus infection, it LOSES its interaction with MAVS and maybe increases interaction with TBK1 (to state this, again, repetitions and quantification is needed). I do not understand why the authors say “THOC7 was recruited to the VISA signalosome after interacting with TBK1” (line 327f). I can also not see why THOC7 “subsequently promoted the polyubiquitination of TBK1” (line 328)– why can this not have happened independent of recruitment to the VISA signalosome? I do not see any evidence for the suggested order of events and strongly advise the authors to carefully remove such over-interpretations from the whole manuscript (including the abstract)!
- Figure 4B: upper panel: be careful! If expression plasmids are all CMV-based, this could simply be a competition at the promoter level (we have seen this happen before). Lower panel: this would be the way to go. Is this really only endogenous TBK1, or is it taken from the same experiment as the upper panel, just probed with TBK1 antibody instead of anti-HA? The legend and methods do not tell… I would be happy if only the endogenous would be shown, but then please include the blot of THOC7!
- Figure 4C: quantification (ratio of ubiquitin signal with divided by without THOC7) would be very advisable.
- Discussion, line 339ff: this study does NOT show involvement of the TREX complex in innate immunity! Only of cytosolic THOC7! I would further more strongly suggest to remove the very last sentence (antiviral drug design), which I know is popular but honestly completely superfluous.
- Discussion: the authors find THOC7 to also affect NFkB activation (see also figure 1E). How can that be explained if it only affects TBK1 (which is not involved in NFkB activation)?
Minor comments:
- Please use MAVS instead of VISA, it is the official name. I also used to call it differently upon its discovery, but we all should agree on one name in the end, it MAVS is the HUGO term, not VISA, not Cardif, not IPS-1.
- Paragraph 3.3 begins with quite a bit of introduction. This info can be easily put into the actual Introduction section; I feel it is a bit out of place here…
- Figure 1B is a bit uncommon and might benefit from a bit more details in the legend. What is the color scale, raw luciferase values?
- Figure 1F: it’s “dimer” not “Dimmer”, please correct (also, better say “monomer” not “Mono”)
- Figure 3: it’s “Nucleus” not “Necleus”
- Figure 3C: the authors might want to add one sentence to explain RIG-I-N and IRF3-5D and why overexpression of plain MAVS, TBK1 and IKKe suffice to trigger signalling. For the authors and myself this is obvious, but it may not be to some readers.
- Figure 4C: it is not stated that THOC7 is myc-tagged in this experiment.
Author Response
Thank you for giving us the opportunity to revise our manuscript. We are sincerely grateful to the comments provided by you. According the suggestions, we repeated some original experiments and did additional new experiments. With these revises, we hope our newly submitted manuscript can meet the criteria for publication in Viruses. We thank your serious consideration on this manuscript.
General comment 1: from my own extensive experience, I know how difficult it is to get promoter reporter assays and also qRT-PCR measurements upon SeV infection to be exactly reproducible. SDs of the data shown throughout the manuscript are so tiny that I would just like to ask the authors to double-check and confirm that all data really is from three INDEPENDENT experiments and error bars represent SD.
We double-checked and confirmed that all data of the manuscript, and replenished partial measurements to three times. In order to reduce error, the virus originated from same batch and was useed with same dose. We have revised the error statistics of partial data. Statistical significance of differences between two groups were analyzed using an unpaired Student’s t-test and that of differences among multiple groups were analyzed using one-way analysis of variance with Tukey's post hoc analysis (GraphPad Prism software ). Error bars represent SD.
- General comment 2: it would be advisable to check the THOC7 effect in at least one different cell line. As 293T cells seem to have very low expression of THOC7, the authors might want to try a cell line with higher expression, such as U2-OS or CAPAN-2 (check https://www.proteinatlas.org/ENSG00000163634-THOC7/cell#rna).
We have not preserved the U2-OS and CAPAN-2 cell in laboratory, so we checked the THOC7 effect in MCF7 cell line which exhibited a higher expression of THOC7 in Protein Atlas web. As expected, overexpression of THOC7 in MCF7 cells showed lower activities of IFN-β promoter (Fig. 1C low panel). Also, we detected endogenous TBK1 in the presence or absence of THOC7, and showed THOC7 promotes the degradation of TBK1 (Fig. S1).
- Figure 1F: native PAGE is not convincing- where is the monomeric IRF3? Compare to fig. 2C! Moreover, the interpretation of this single experiment (line 179, 193, abstract, discussion) is not backed by this blot (“THOC7 significantly reduced phosphorylation and dimerization”). This experiment should be repeated and quantified (n=3).
We repeated the experiments to three times, and detected normal monomeric IRF3 in Fig. 1F. In previous similar experiments, we detected weak monomeric IRF3 and strong band of dimer IRF3occasionally. While this phenomenon cannot be explained, we infer that it is the reason of sodium deoxycholate and different batches of IRF3 antibody. But we still maintained that the experiment results have reference value. The quantification was added in Figure 1F (right panel).
- Figure 2: panel B seems to show the quantification of the blot in panel A across three repetitions. This is great and exactly what I would suggest to show also for the other blots (see comment above). However, there is not legend text for panel B- please also state how western blots were quantified!
We added the method of quantification in section of Materials and Methods (2.4). Quantification of Western blotting bands from three experiments was performed with Image J.
- Figure 2: please explain why endogenous protein was not looked at? Was protein expression too low? There should be an antibody available!
A work well antibody of THOC7 was not found in our previous experiments. We purchased THOC7 antibody from BBI life science (Cat. No. D121872). The antibody only could detect overexpression THOC7 but endogenous THOC7. AS shown in Fig.S1, the antibody also could not detect endogenous THOC7 in MCF7 cell line. The data of THOC7 q-PCR in Fig.2B also act as a supplementary proof to the role of RNAis.
- Figure 2E: again, monomeric IRF3 is not showing up, is there an explanation for this? Furthermore, please repeat to a total of n=3 and include error bars in the quantification (panel E right).
We detected normal monomeric IRF3 in Fig. 2E. The error bars were added across three experiments.
- Figure 2F: quantify, see above!
It has been corrected.
- Figure 3: see “general comment 1” (n and SD)
It has been double-checked and corrected.
- Figure 4A: I cannot follow the interpretations of this experiment! To me, it seems very clear that THOC7 constitutively interacts with MAVS and IKKe, possibly a little bit also with TBK1. After virus infection, it LOSES its interaction with MAVS and maybe increases interaction with TBK1 (to state this, again, repetitions and quantification is needed). I do not understand why the authors say “THOC7 was recruited to the VISA signalosome after interacting with TBK1” (line 327f). I can also not see why THOC7 “subsequently promoted the polyubiquitination of TBK1” (line 328) – why can this not have happened independent of recruitment to the VISA signalosome? I do not see any evidence for the suggested order of events and strongly advise the authors to carefully remove such over-interpretations from the whole manuscript (including the abstract)!
We agree with the opinion of review. The interaction of TBK1 and THOC7 was increased after virus infection, and the effect was also observed in Fig.1A. It has been corrected in manuscript.
- Figure 4B: upper panel: be careful! If expression plasmids are all CMV-based, this could simply be a competition at the promoter level (we have seen this happen before). Lower panel: this would be the way to go. Is this really only endogenous TBK1, or is it taken from the same experiment as the upper panel, just probed with TBK1 antibody instead of anti-HA? The legend and methods do not tell… I would be happy if only the endogenous would be shown, but then please include the blot of THOC7!
All plasmids are indeed CMV-based and the lower panel was only endogenous TBK1.The up panel and lower panel were two independent experiments. Though a concern of a competition at the promoter level, we also maintained that the experiment result (up panel) have reference value. Additionally, we check the effect in MCF7 cell, and knock down of THOC7 restored the TBK1 protein level.
- Figure 4C: quantification (ratio of ubiquitin signal with divided by without THOC7) would be very advisable.
The experiments were repeated to three times, and quantification was performed by Image J.
- Discussion, line 339ff: this study does NOT show involvement of the TREX complex in innate immunity! Only of cytosolic THOC7! I would further more strongly suggest to remove the very last sentence (antiviral drug design), which I know is popular but honestly completely superfluous.
It has been corrected.
- Discussion: the authors find THOC7 to also affect NFkB activation (see also figure 1E). How can that be explained if it only affects TBK1 (which is not involved in NFkB activation)?
We added discuss on this point in line 339. We doubt that THOC7 suppresses NF-κB activation probably by targeting canonical IKKs which possess sequence homology with TBK1.
Minor comments:
- Please use MAVS instead of VISA, it is the official name. I also used to call it differently upon its discovery, but we all should agree on one name in the end, it MAVS is the HUGO term, not VISA, not Cardif, not IPS-1.
It has been corrected.
- Paragraph 3.3 begins with quite a bit of introduction. This info can be easily put into the actual Introduction section; I feel it is a bit out of place here…
For reviewer or other readers who good understood of this field, this may be obvious, but it may not be to other readers. The information can help readers understand of the following experimental design better. If we put it into the Introduction section, we think it may disrupt the original structure of introduction.
- Figure 1B is a bit uncommon and might benefit from a bit more details in the legend. What is the color scale, raw luciferase values?
We think Figure 1B is more intuitive and visual to reflect the function of THOC7 ininitial function screening. We added details in the legend.
- Figure 1F: it’s “dimer” not “Dimmer”, please correct (also, better say “monomer” not “Mono”)
It has been corrected.
- Figure 3: it’s “Nucleus” not “Necleus” .
It has been corrected.
- Figure 3C: the authors might want to add one sentence to explain RIG-I-N and IRF3-5D and why overexpression of plain MAVS, TBK1 and IKKe suffice to trigger signalling. For the authors and myself this is obvious, but it may not be to some readers.
We added the interpretation in line 247.
- Figure 4C: it is not stated that THOC7 is myc-tagged in this experiment.
We added myc-THOC7 in section of Materials and Methods (2.2. Plasmids).
Reviewer 2 Report
He et. al. describe THO complex subunit 7 homolog negatively regulates type I Interferon response by interacting TBK1. The work is interesting and the conclusions are generally accurate and supported by experiments. The manuscript reads well.
I have only a few questions/comments that need to be addressed.
1. In Fig. 1C, they showed the reporter activity to show the interferon response. It would be helpful to show the change of transcript level of genes expressed by ISRE or IFN promoter using quantitative PCR.
2. In Fig. 1F, 2C, and 2F, authors described that IRF-3 dimerization and phosphorylation were inhibited by THOC7 overexpression or exerted opposite effects by knockdown. As shown in Fig 2E right panel, the quantitative analysis of IRF3 dimerization or phosphorylation to actin would be more helpful to understand.
3. In Fig. 4B, authors claimed that THOC7 promotes the proteasomal degradation of TBK1. The data on the inhibition of degradation of TBK1 using the proteasome inhibitor such as MG132 would be more supportive.
4. Minor comments;
Line 344 : ‘Cellular’ needs to be changed as cellular.
Author Response
Thank you for giving us the opportunity to revise our manuscript. We are sincerely grateful to the comments provided by you. According the suggestions, we repeated some original experiments and did additional new experiments. With these revises, we hope our newly submitted manuscript can meet the criteria for publication in Viruses. We thank your serious consideration on this manuscript.
He et. al. describe THO complex subunit 7 homolog negatively regulates type I Interferon response by interacting TBK1. The work is interesting and the conclusions are generally accurate and supported by experiments. The manuscript reads well.
I have only a few questions/comments that need to be addressed.
1. In Fig. 1C, they showed the reporter activity to show the interferon response. It would be helpful to show the change of transcript level of genes expressed by ISRE or IFN promoter using quantitative PCR.
We detected transcript level of ifnb1 and isg56 using quantitative PCR. The data indicated that THOC7 overexpression resulted in lower expression of IFN-β, and ISG56 (Fig. 1G).
2. In Fig. 1F, 2C, and 2F, authors described that IRF-3 dimerization and phosphorylation were inhibited by THOC7 overexpression or exerted opposite effects by knockdown. As shown in Fig 2E right panel, the quantitative analysis of IRF3 dimerization or phosphorylation to actin would be more helpful to understand.
We added the quantitative analysis in Fig. 1F, 2C, and 2F. And the method of quantification was described in materials and methods.
3. In Fig. 4B, authors claimed that THOC7 promotes the proteasomal degradation of TBK1. The data on the inhibition of degradation of TBK1 using the proteasome inhibitor such as MG132 would be more supportive.
The degradation of TBK1 was usually carried out through the proteasome pathway and could be inhibited by the proteasome inhibitor MG132. To investigate the mechanism, MG132 were supplemented in the degradation effect without newly synthesized protein (CHX treatment). The result showed that, MG132 could efficiently restore the TBK1 protein level in the presence of THOC7 (Fig. 4C).
4. Minor comments;
Line 344 : ‘Cellular’ needs to be changed as cellular.
It has been corrected.
Round 2
Reviewer 1 Report
In the revised manuscript, most of my initial points of concern have been addressed. I have only a few further remarks:
general comment 1: OK; please state clearly in the legends that n=3 from independent experiments
general comment 2: good to have MCF7 cells included now, great improvement! Just: in Fig. S1, I cannot see a downregulation of TBK1 by THOC7 expression, only by THOC7 + Sendai. This should be corrected in the text.
Figure 1F: nice to see monomeric IRF3 now. BUT: it seems this experiment has been repeated, however the western blot panels for p-IRF3 and IRF3 have NOT been changed from the first submission. This means this figure is not from one experiment, but it should be! Please correct!
Figure 2E: same as 1F! Native panel was exchanged, denaturing ones not. Please use blots from one experiment!
Figure 3: OK. Please give the n in the legend.
Figure 4A: I can see the improvement, but still I cannot completely follow the authors' arguments. Note that the interaction between THOC7 and MAVS in in fact DECREASING upon virus infection, so how can one say THOC7 is recruited to the MAVS signalosome upon virus infection? To me, still, the MAVS interaction and the TBK1 interaction are not necessarily linked! At least I don't see this in the data. As for the new experimetns in figure 4 in this context: for Fig. S1, see my comments above. For the CHX/MG132 experiments (nice to see them included!), I wonder wether there is a little mistake in the quantification panel: the very right data set reads "MG132", should this not be "CHX"? Compare to the blots above. Also, please include in the figure a legend to describe the black and white bars (I assume it's with and without THOC7).
Lastly, please add one more sentence in the methods section as to how the western blots were quantified: did you use film and then scanned it to quantify or ECL imaging? In the latter case, which device was used?
Author Response
Dear reviewer,
Thank you for your precious comments on our manuscript entitled “THO Complex Subunit 7 Homolog Negatively Regulates Cellular Antiviral Response against RNA Viruses by Targeting TBK1” (Manuscript ID: viruses-413355) . The cover letter to is attached. Thank you.
In the revised manuscript, most of my initial points of concern have been addressed. I have only a few further remarks:
general comment 1: OK; please state clearly in the legends that n=3 from independent experiments
It has been added to the legends.
general comment 2: good to have MCF7 cells included now, great improvement! Just: in Fig. S1, I cannot see a downregulation of TBK1 by THOC7 expression, only by THOC7 + Sendai. This should be corrected in the text.
It has been corrected in the text (line 296).
Figure 1F: nice to see monomeric IRF3 now. BUT: it seems this experiment has been repeated, however the western blot panels for p-IRF3 and IRF3 have NOT been changed from the first submission. This means this figure is not from one experiment, but it should be! Please correct!
It has been corrected in the new submission.
Figure 2E: same as 1F! Native panel was exchanged, denaturing ones not. Please use blots from one experiment!
It has been corrected in the new submission.
Figure 3: OK. Please give the n in the legend.
It has been added to the legends.
Figure 4A: I can see the improvement, but still I cannot completely follow the authors' arguments. Note that the interaction between THOC7 and MAVS in in fact DECREASING upon virus infection, so how can one say THOC7 is recruited to the MAVS signalosome upon virus infection? To me, still, the MAVS interaction and the TBK1 interaction are not necessarily linked! At least I don't see this in the data. As for the new experimetns in figure 4 in this context: for Fig. S1, see my comments above. For the CHX/MG132 experiments (nice to see them included!), I wonder wether there is a little mistake in the quantification panel: the very right data set reads "MG132", should this not be "CHX"? Compare to the blots above. Also, please include in the figure a legend to describe the black and white bars (I assume it's with and without THOC7)
All concerning the interaction of THOC7 with MAVS signalosome have been corrected in whole manuscript. X-axis description in Fig4C lower panel have changed “MG132” into “CHX”, and legend have added to Fig4C and other new figures to describe the black and white bars.
Lastly, please add one more sentence in the methods section as to how the western blots were quantified: did you use film and then scanned it to quantify or ECL imaging? In the latter case, which device was used?
It has been added to the the methods section (in 2.4).
Reviewer 2 Report
The authors have addressed my comments.
One minor point is that
x-axis description in Fig4C lower panel should be changed
"MG132" into "CHX".
Author Response
Dear reviewer,
Thank you for your precious comments on our manuscript entitled “THO Complex Subunit 7 Homolog Negatively Regulates Cellular Antiviral Response against RNA Viruses by Targeting TBK1” (Manuscript ID: viruses-413355) . The mistake about Fig4C has been corrected. Thank you.
Reviewer #2:
- One minor point is that x-axis description in Fig4C lower panel should be changed "MG132" into "CHX".
- It has been corrected in the new submission.